# Digital solutions to follow up on discharged new parents—A systematic literature review

Zada Pajalic[1]*, Alona Rauckiene[2], Grethe Savosnick[1], Irena Bartels[3], Jean Calleja-Agius[4], Diana Saplacan[5], Sigríður Sía Jónsdóttir[6], Babak Asadi-Azarbaijani[1]

1 Faculty of Health Sciences, VID Specialized University, Oslo Norway, 2 Department of Health Research and Innovation Science Centre, Klaipeda University, Klaipeda Lithuania, 3 East Tallinn Central Hospital, Tallinn Estonia, 4 Department of Anatomy, Faculty of Medicine and Surgery, University of Malta, Valletta Malta, 5 Department of Informatics, University of Oslo, Oslo Norway, 6 School of Health Science, University of Akureyri, Akureyri Iceland

* Zada.pajalic@vid.no

**Data Availability Statement:** All data are in the manuscript and/or supporting information files.

**Funding:** The authors received no specific funding for this work.

## Abstract

New parents and their newborns are followed up after discharge either through home visits from midwives/nurses or using information and communication technology. This follow-up focuses on individual needs related to breastfeeding and infant feeding, practical advice on caring for babies, supporting and strengthening the new mother's knowledge and self-confidence concerning child development and parenting skills, and supporting the relationship between parents and baby. This systematic review aims to integrate available research results that describe new parents' experiences when health and care providers used telemedicine as a platform for follow-up after discharge from the childbirth department. This literature review was conducted following the PRISMA statement and was prospectively registered in PROSPERO CRD42021236912. The studies were identified through the following databases: AMED, Academic, EMBASE, Google Scholar, Ovid MEDLINE via PubMed, Cochrane database, and CINAHL. Results from these studies were compiled using thematic analysis. A total of 886 studies were identified. Screening resulted in eight studies that met the inclusion criteria. Thematic analysis produced the following themes: a) *Flexibility and convenience of digital support*, b) *Digital literacy*, c) *Parents feeling safe with digital support*, and d) *Adequate substitute for physical meetings*. New parents who live in a home environment with a relaxed atmosphere and around-the-clock digital support experience a sense of control, security, full attention, and encouragement. Digital follow up at home has proven effective because it can meet the support needs of new parents when necessary.

## Author summary

Since introducing information and communication technology in the health and care sector, it has been used on trial to follow up with new parents and their newborns. The idea behind follow-up using information and communication technology is to offer round-the-clock advice on breastfeeding and infant feeding and practical advice on baby care to

**Competing interests:** The authors have declared that no competing interests exist.

support and strengthen the new parents' knowledge of the child's development. Here, we use a systematic literature review to determine how new parents experienced this type of follow-up. We found that follow-up with the information and communication technology was perceived as flexible. Most new parents have digital skills, which makes them feel safe with digital support, and this type of follow-up is perceived as an adequate substitute for physical meetings. Our study provides new insights into digital monitoring as a new way to meet the needs of new parents.

## 1.0 Introduction

Parity, birth-related factors, and psychosocial predictors factors indicate when it is appropriate for a new mother to be discharged and involved in the organisation of postnatal care. Discharge is when a healthy mother and baby return home from hospital after a birth [1]. Parity, birth-related factors, and psychosocial predictors indicate when a new mother can be discharged and involved in the organisation of postnatal care [2]. Family bonding, which is vital for the whole family, can be achieved more rapidly if the family is together at home and if partners are involved from birth [3].

Potential advantages of early discharge for new parents and their babies include a peaceful home environment, positive impact on breastfeeding, enhanced emotional well-being, minimised exposure to infection risks, and increased parental relaxation and involvement in child care [4]. Follow-up of new parents after discharge is usually offered by the midwife [5] via home visits [6] or telemedicine [7]. This follow-up often focuses on individual needs related to breastfeeding and infant feeding, practical advice on caring for babies, supporting and strengthening the new mother's knowledge and self-confidence, child development, parenting skills, and supporting the mother–child relationship [8].

Telemedicine is an umbrella term for the use of telecommunications technology to provide medical support at a distance [9]. Telemedicine is divided into different categories based on the interaction between users and the type of information used. The interaction can take place in real-time (synchronous) or be pre-recorded (asynchronous), and the information can be transmitted between users via audio, text, images, and videos [9]. A similar concept to telemedicine is eHealth, a term that comprises health service delivery between healthcare professionals and patients through the internet and related technologies [10]. Telemedicine offers new possibilities and enables support for many patients remotely in a cost-effective way [11].

There is an international trend in welfare countries to shorten the postpartum length of stay in hospitals, driven by cost containment and hospital bed availability [12]. The average postpartum stay after vaginal delivery varies between countries [1]. Discharge after childbirth can be challenging for new parents, many of whom have a great need for practical support and expert guidance. As such, early discharge requires integrated care services, 24-hour availability of qualified healthcare professionals for advice and support, home visits and the opportunity for home support from qualified healthcare professionals [13]. Moreover, feeling safe following an early return home depends on whether parents make that decision themselves and whether breastfeeding has commenced [14]. Research shows that parents perceive discharge as positive when they receive adequate follow-up and help from healthcare professionals and know who to contact if needed [13]. Parents experience early discharge after birth as successful if they a) receive follow-up from a midwife and home visits after they return home, with the opportunity to obtain answers to questions when they arise; b) receive sufficient information; and c) are able to form a trusting relationship with the midwife [15].

Postpartum care must be based on a woman's decisions, clinical indications, family preferences, and care context [16]. Planned follow-up care has a significant impact on the health and well-being of parents, children, and families both in the short and the long term [17,18]. Research shows that home-based postnatal care is well accepted by new parents who are discharged after childbirth [19]. Moreover, when parents and partners feel empowered by their midwife, this enhances feelings of self-confidence and a sense of security after discharge from a maternity unit. All postpartum interventions related to the provision of continuity of care (quality of care over time) are associated with satisfaction and impact the family's psychological and social outcomes [16]. With this review, we want to increase our knowledge of new parents' subjective perceptions, evaluations and interpretations regarding digital follow-up from the care provider's side after discharge from the maternity ward. The new understanding will give us insight into what is perceived to be working well and what challenges remain.

"This systematic review aims to integrate available research results that describe parents' experiences when health and care providers used telemedicine as a platform for follow-up after discharge from the childbirth department".

## 2.0 Methods

The systematic review was performed following the PRISMA statement and was prospectively registered in PROSPERO [20] (no. CRD42021236912).

### 2.1 Data sources and search strategy

The PROSPERO database was searched to determine whether similar studies had been completed or were still in progress. We could not find any registered studies with the same aim as ours.

### 2.2 Eligibility criteria

The review aimed to search for peer-reviewed primary qualitative studies that responded to the study's objective. Studies with a qualitative design describe parents' experiences, which answers the purpose of the study. The searches were performed between 030121–311221. A new search was carried out in October 2022. No new studies meeting the inclusion criteria were found.

### 2.3 Inclusion criteria

- All types of telemedicine used by healthcare professionals to follow-up discharged new parents from the childbirth department
- New parents discharged to their own homes
- Studies focusing on support in private homes
- All digital delivery formats including web and app based
- Qualitative studies
- Welfare countries

### 2.4 Exclusion criteria

- Usual medical follow-up (e.g. return visits to hospital, home visits, follow-up at care centers)

- Not target population

- Not primary study

- Protocol, review articles

- Books

- Studies in hospitals

- Quantitative studies

### 2.5 Information sources

At VID Specialised University in Oslo, Norway, a science librarian and first author (ZP) identified primary studies in the following bibliographic electronic databases: AMED, Academic, EMBASE, Ovid MEDLINE via PubMed, Google Scholar, Cochrane database and CINAHL.

### 2.6 Search strategy

Systematic searches were conducted for studies published in English, Norwegian, and Swedish. The searches were limited to the period between 2015 and 2022 to include the newest studies. The search strategy incorporated the following MeSH keywords singly and in combination with each other (combined with the Boolean operators OR and AND): patient, birth, early, discharge, postpartum, experience, length of stay, parent, semi-structured interviews, cross-sectional, sense of security, follow-up, health system, primipara, multipara, home-based postnatal care, midwife, home visits, ambulatory care, health service, managed care, aftercare, self-care, fear of infection, telemedicine, expectations (Fig 1).

### 2.7 Study selection and review process

In total, 886 studies were identified through literature searches and were first imported into the EndNote reference manager [21] and then into Rayyan [22] (a web tool for systematic reviews that aids in the comparison of decisions regarding inclusion or exclusion of studies). All duplicates (n = 136) were removed resulting in 750 publications. Titles and abstracts for the 750 publications were then screened according to inclusion and exclusion criteria. Of these, 58 were screened in full text, 15 of which were assessed for quality using the Critical Appraisal Skills Programme (CASP) [23] checklist for qualitative studies. CASP assessment ranges from 0–10. Ten points mean that the study is of high quality. Of these 15 publications, 8 [24–31] were judged to have the best combination of relevance and quality (Fig 2, Tables 1 and 2).

Screening of titles and abstracts was performed by all co-authors, first independently and then together. In the case of conflicting independent decisions, the authors reviewed the titles again together and discussed them to reach consensus. Full text screening and CASP quality review were performed by all authors. The final decision of which studies to include was made by all reviewers.

### 2.8 Data synthesis

We chose to compile results (extracted data) from the selected articles using thematic synthesis, following Braun and Clarke [32]. We have included and used all extracted data throughout the analysis process step by step from the identification of initial codes and grouping the codes into themes as described below.

01.03.21
Search 1: ((((((((((((((("new mother")) OR ("new parents")) OR (birth)) OR (Childbirth)) OR (primiparas)) OR (multiparas)) OR (parent)) OR ("new parent")) OR ("new parents")) OR ("home based postnatal care")) OR (postpartum)) OR ("postnatal care")) OR ("managed care")) OR ("Ambulatory care")) OR ("After care")) OR ("home visit")) OR ("home visits")
**Number: 1 032 717**Search 2: (((((((((("health system") OR (telemedicine)) OR ("Health services")) OR ("mobile health")) OR ("Health, mobile")) OR (mhealth)) OR (telehealth)) OR (ehealth)) OR (telecommunication)) OR (telenursing)
**Number: 687 119**

Search 3: ((((("early postnatal discharge") OR ("early discharge")) OR ("length of stay")) OR ("very early discharge")) OR ("early postpartum discharge")) OR ("early patient discharge")
**Number: 127 601**

Search4: ((((("semistructured interview") OR ("semistructured interviews")) OR ("cross sectional")) OR ("content analysis")) OR (qualitative)) OR ("grounded theory")
**Number: 833 989**

Search 5: #1 AND #2 AND #3
(((((((((((((((("new mother")) OR ("new parents")) OR (birth)) OR (Childbirth)) OR (primiparas)) OR (multiparas)) OR (parent)) OR ("new parent")) OR ("new parents")) OR ("home based postnatal care")) OR (postpartum)) OR ("postnatal care")) OR ("managed care")) OR ("Ambulatory care")) OR ("After care")) OR ("home visit")) OR ("home visits")) AND (((((((((("health system") OR (telemedicine)) OR ("Health services")) OR ("mobile health")) OR ("Health, mobile")) OR (mhealth)) OR (telehealth)) OR (ehealth)) OR (telecommunication)) OR (telenursing))) AND ((((((("early postnatal discharge") OR ("early discharge")) OR ("length of stay")) OR ("very early discharge")) OR ("early postpartum discharge")) OR ("early patient discharge"))) AND ((((((("semistructured interview") OR ("semistructured interviews")) OR ("cross sectional")) OR ("content analysis")) OR (qualitative)) OR ("grounded theory"))
**Number: 98**

Search 6: #5 AND English AND from 2015/1/1 - 2021/12/31
(((((((((((((((("new mother")) OR ("new parents")) OR (birth)) OR (Childbirth)) OR (primiparas)) OR (multiparas)) OR (parent)) OR ("new parent")) OR ("new parents")) OR ("home based postnatal care")) OR (postpartum)) OR ("postnatal care")) OR ("managed care")) OR ("Ambulatory care")) OR ("After care")) OR ("home visit")) OR ("home visits")) AND (((((((((("health system") OR (telemedicine)) OR ("Health services")) OR ("mobile health")) OR ("Health, mobile")) OR (mhealth)) OR (telehealth)) OR (ehealth)) OR (telecommunication)) OR (telenursing))) AND ((((((("early postnatal discharge") OR ("early discharge")) OR ("length of stay")) OR ("very early discharge")) OR ("early postpartum discharge")) OR ("early patient discharge"))) AND ((((((("semistructured interview") OR ("semistructured interviews")) OR ("cross sectional")) OR ("content analysis")) OR (qualitative)) OR ("grounded theory")) Filters: English, from 2015/1/1 - 2021/12/31
**Number47**

**Fig 1. Example searches history in database PubMed.**

Analysis of selected study results was performed by BAA and ZP and discussed with all co-authors until consensus was reached.

The first phase of our work was to extract the results from the selected studies in a separate document. After reading the text several times, we identified initial codes, which we then grouped into different themes. All co-authors critically reviewed the potential themes using a process of dialogue and consensus. Finally, we defined and named the following themes: *Flexibility and convenience of digital support*, b) *Digital literacy*, c) *Parents feeling safe with digital support*, and d) *Adequate substitute for physical meetings*.

## 3.0 Results

### 3.1 Flexibility and convenience of digital support

This study showed that the sharing of information between parents and healthcare professionals was essential [24–31]. Various digital platforms provided a combination of asynchronous

PRISMA flowchart of study selection procedure

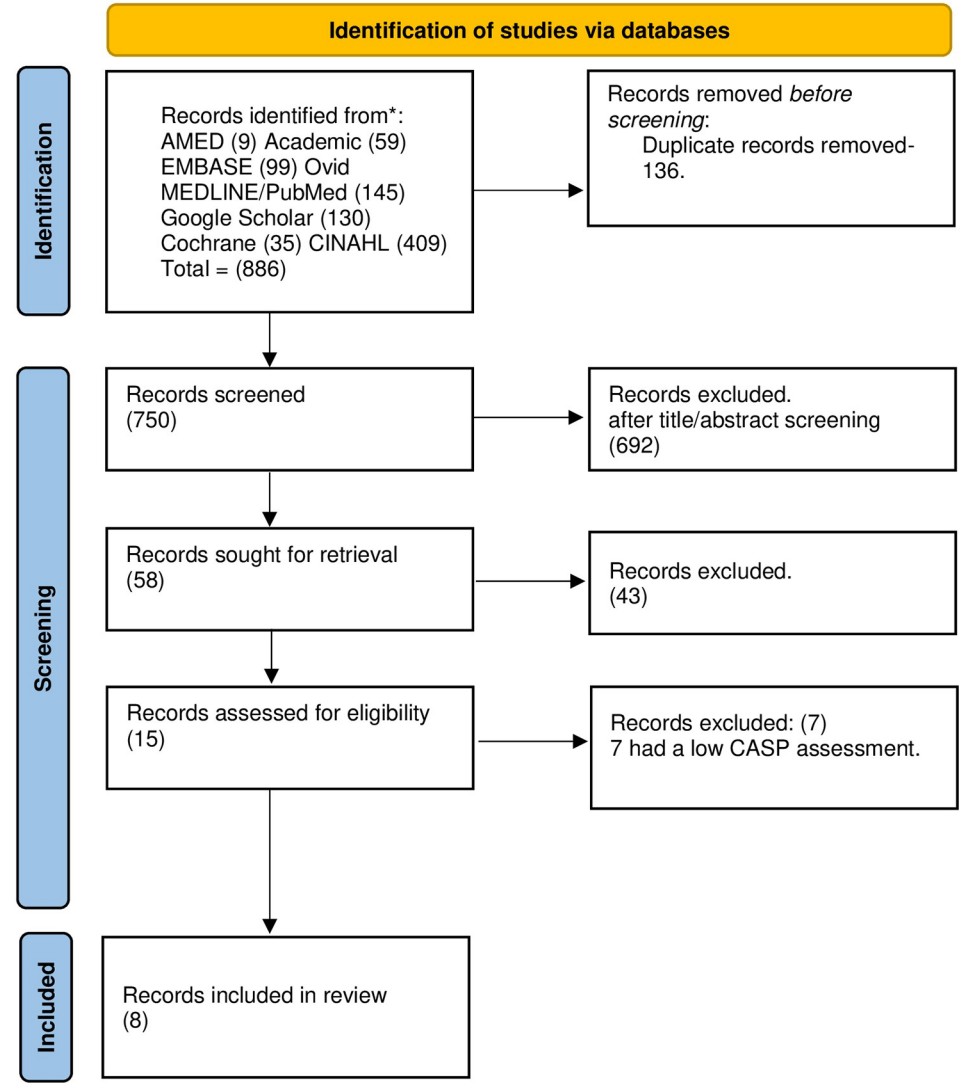

**Fig 2. Supporting information files S1 Checklist.**

(chat bots) and synchronous (staff respond directly via audio or video) methods to facilitate communication with parents out of hospital. Asynchronous communication offered an accessible way to obtain help after early discharge from maternity units. New parents appreciated having this information available, as well as an online chat option. This was also an excellent way to refresh the knowledge of those who already had children. Most parents found the new communication methods for contacting healthcare professionals easy to use and that they conveyed a range of information effectively, thus saving time. All parents felt safe because they had

**Table 1. CASP assessment of included studies\*.**

| Study | Q1 | Q2 | Q3 | Q4 | Q5 | Q6 | Q7 | Q8 | Q9 | Q10 | Sum |
|---|---|---|---|---|---|---|---|---|---|---|---|
| Lindberg et al. 2009 | 1 | 1 | 1 | 1 | 1 | 1 | 1 | 1 | 1 | 1 | 10 |
| Danbjørg et al. 2014 | 1 | 1 | 1 | 1 | 1 | 1 | 1 | 1 | 1 | 1 | 10 |
| Danbjørg et al. 2014 | 1 | 1 | 1 | 1 | 1 | 1 | 1 | 1 | 1 | 1 | 10 |
| Danbjørg et al. 2015 | 1 | 1 | 1 | 1 | 1 | 1 | 1 | 1 | 1 | 1 | 10 |
| Garne et al. 2016 | 1 | 1 | 1 | 1 | 1 | 1 | 1 | 1 | 1 | 1 | 10 |
| Shorey et al. 2018 | 1 | 1 | 1 | 1 | 1 | 1 | 1 | 1 | 1 | 1 | 10 |
| Garne Holm et al. 2019 | 1 | 1 | 1 | 1 | 1 | 1 | 1 | 1 | 1 | 1 | 10 |
| Shorey et al. 2019 | 1 | 1 | 1 | 1 | 1 | 1 | 1 | 1 | 1 | 1 | 10 |

\*CASP Checklist (Yes = 1; Can't tell = X; No = 0)

10 questions to help you make sense of qualitative research: Q1 Was there a clear statement of the aims of the research? Q2 Is a qualitative methodology appropriate? Q3 Was the research design appropriate to address the aims of the research? Q4 Was the recruitment strategy appropriate to the aims of the research? Q5 Was the data collected in a way that addressed the research issue? Q6 Has the relationship between researcher and participants been adequately considered? Q7 Have ethical issues been taken into consideration? Q8 Was the data analysis sufficiently rigorous? Q9 Is there a clear statement of findings? Q10 How valuable is the research?

access to the information they needed. It should be noted, however, that in terms of parents' ability to find and retrieve information, digital literacy was crucial.

This review also identified 24-hour access to information and the ability to access the same information repeatedly as additional positive aspects. Digital platforms provided the opportunity to distribute information based on current needs so that even new parents could share their knowledge and experiences via chat functions. Written information proved to be easily accessible around the clock, whether through an application (app), chat, or short text messages. Some groups also had access to web pages containing answers to the most frequently asked questions. Digital platforms created and led by midwives were perceived as the most credible sources for parents. Being able to receive rapid answers to questions that arose throughout the day or night made several new parents feel closer to their midwives and that they were being given the midwives' full attention whenever they needed it. Access to information thus gave parents a sense of control, security, and reassurance.

In addition, it was important for most new parents to receive positive feedback and confirmation from healthcare professionals. The flexibility that digital communication with healthcare professionals offered gave the parents a feeling of control and peace. New parents also stated that it was easier and more convenient to communicate digitally, and that sending a message was less disruptive. Some participants preferred asynchronous video instruction rather than face-to-face guidance from healthcare professionals because they could watch the videos whenever they had time and view them repeatedly if desired. In one of the studies, the visual contact established via videoconference facilitated a warm relationship between parents and healthcare professionals. The parents could see the nurses acknowledging their facial expressions and body language during their conversations when parents shared their observations related to their babies with the nurses. In this way, parents felt they were receiving the full attention of the healthcare professionals.

## 3.2 Digital literacy

Another theme that emerged was digital literacy among parents. Some study participants expressed that they were both surprised by and unprepared for early discharge after delivery; indeed, some had negative experiences because of this and had felt unwelcome [24–31]. Many of the new parents in the studies would have chosen to stay longer in hospital, with the main

**Table 2. Overview of included studies.**

| Author, year, country | Aim | Welfare technology used | Method design | Participants | Data collection | Data analysis | Results |
|---|---|---|---|---|---|---|---|
| Linberg et al. 2009 Sweden | To describe parents' experiences of using videoconferencing when discharged early from a maternity unit | Videoconferencing | Combination of quantitative and qualitative design | 9 couples | Questionnaires and interviews | Thematic content analysis | Study results showed that parents were confident regarding using the technology, being face-to-face during the videoconference (VC), having control over their privacy, and feeling confident especially 'when worries and concerns were met, and answers were received'. Using VC as a support in cases of early discharge after childbirth can facilitate a meeting that enables new parents to be guided by the midwife in their transition to parenthood. |
| Danbjørg et al. 2014 Denmark | To describe the process of the design, development, and testing of an application as a viable information technology solution | Application | Participatory design with in-depth consultation with users | 9 parents & 1 father | Participant observation, field notes, individual telephone interviews | Systematic text condensation | The families found it natural to communicate online, and they did not feel any barriers. The app met the new families' needs for follow-up support. The testing suggests that the new families and the nurses found the app viable, but the app requires refinements and wider testing. |
| Danbjørg et al. 2014 Denmark | To identify the nursing support needs of new parents and their infants during the first seven days postpartum, by drawing on the experiences of all stakeholders in early postnatal discharge from hospital, and thereby gaining new knowledge to investigate further whether telemedicine is a viable option in providing the required support | Telemedicine | Participatory and qualitative design | 19 parents | Participant observation, interviews, workshop | Systematic text condensation | Families that were discharged early requested more availability from the healthcare system to respond to their concerns and questions during the postnatal period. They asked for new ways to communicate that would meet their needs for more individualised and timely information and guidance. Study results indicate that it may be possible to meet the follow-up support needs of these families through new ways to communicate, such as online communication. |
| Danbjørg et al. 2015 Denmark | To explore how parents experienced the use of telemedicine following early postnatal discharge | Application | Participatory and qualitative design | 27 parents & 11 partners | Interviews | Systematic text condensation | Parents were confident in using the application (app) and found it natural to seek information and communicate online; they did not experience any barriers in contacting the nurses using asynchronous communication. The functionalities of the app (i.e., chat, knowledge base and automated messages) met the needs of the new parents and induced a sense of security and parental self-efficacy. |

(*Continued*)

**Table 2.** (Continued)

| Author, year, country | Aim | Welfare technology used | Method design | Participants | Data collection | Data analysis | Results |
|---|---|---|---|---|---|---|---|
| Garne et al. 2016 Denmark | To identify parents' needs when seeking to provide neonatal home care supported by telemedicine | Telemedicine | Participatory and qualitative design | 19 parents | Observational studies, individual interviews, and focus group interviews | Systematic text condensation | This study supports the use of telemedicine for neonatal home care because it gives parents the feeling of being a family and promotes their self-efficacy. Telemedicine-provided home care gives parents the guidance they need from nurses when they do not require hands-on support. Parents also outlined the need for a technological 'bell cord', such as videoconferencing, timely e-mail communication, and a knowledge base of information regarding infant nutrition and breastfeeding. |
| Shorey et al. 2018 Singapore | To explore the views of parents of newborns with regard to the content and delivery of a mobile health (mHealth) app-based postnatal educational programme | Application | Qualitative design | 17 participants (5 couples, 4 partners, and 3 parents) | Interviews | Thematic content analysis | Study participants reported that mobile health app-based support was a good source of information that was tailored to the local context. The parents assessed the facilitator of the featured communication platform, a midwife, as providing trustworthy advice. Belonging to a virtual community beyond the hospital gave the parents the feeling that they were not alone and were supported by healthcare professionals. |
| Garne Holm et al. 2019 Denmark | To explore parents' experiences with neonatal tele-homecare (NTH) | Application | Qualitative design | 49 parents | Interviews | Systematic text condensation | The results from the study identified that NTH could facilitate family-centred care (FCC). The telehealth service served as a personal lifeline to clinical expertise, which was easily accessed. Through the videoconference sessions, the parents felt acknowledged regarding their parenting skills and their observations of their infants. Study findings indicate that tele-homecare can be a supportive practice for parents. |
| Shorey et al. 2019 Singapore | To examine the experiences and perceptions of participants in a supportive education parenting programme intervention study | Application | Qualitative design | 16 parents (6 control and 10 intervention) | Interviews | Thematic content analysis | Parents from the intervention group reported having good experiences with receiving sufficient mobile health support. The multifeatured, technology-based intervention was effective in improving parental outcomes and was well received by parents. |

reasons being the need for breastfeeding support and help interpreting newborns' signals. However, the ability to use digital technology once they arrived home enhanced parents' feelings of security and gave them a sense of control.

All of the parents in the studies rapidly oriented themselves by using the media offered by the hospital as a support at home. Digital platforms with direct or asynchronous communication provided when needed reduced parents' feelings of insecurity about their new role. Asynchronous provision of information also helped meet the need for more frequent support. In addition, new parents reported using digital platforms (e.g. social media) as an alternative support strategy, seeking information on their own if they felt they were receiving insufficient support, were uncertain about something, or questioned the advice given. Some participants even wanted to compare the information they were receiving.

The most pressing need was to have more accessible personalised information on digital platforms. Every task that the parents accomplished via digital support strengthened their self-confidence and helped create a positive experience. Their self-confidence grew alongside their increased understanding of their baby's behaviour. Another freedom that the parents discovered with digital support was that they could ask any questions they desired, without feeling ashamed, which they found liberating. In step with increased self-confidence, some new parents also chose to share their experiences with other parents through this platform because they felt confident.

One interesting finding is that, in some cases, there was a mismatch in digital competence between healthcare professionals and parents, with healthcare professionals being at a disadvantage. When it became evident that the parents had better digital skills than the midwives, this led to a disparity in knowledge of how to navigate digital platforms between new parents and healthcare professionals. Some parents appreciated that they could help the midwives improve their digital competence via interactive collaboration; they felt confident regarding their capacity and ability to use the app or improve healthcare professionals' digital skills. By establishing some control, in addition to helping them feel prepared and strong in their new role, this kind of mutual knowledge exchange became a way to further empower parents.

### 3.3 Parents feeling safe with digital support

The results showed that digital follow-up met the parents' need for emotional support and gave them a sense of security [24–31]. Several parents reported that the digital follow-up was better than they expected to be. It was essential for participants to know that they could manage everything themselves and feel safe in their new role as parents.

For the parents, the most positive aspect was being able to communicate directly with the healthcare professionals and feeling comfortable and natural in front of the web camera. This contributed to a sense of control and made it easier to absorb information. Many likened the digital follow-ups at home to a digital lifeline. Indeed, new parents who could obtain the information they needed at home reported improved emotional well-being. They felt safe and relaxed, found it easier to seek help, and had more self-confidence when performing routine infant care tasks. It was important for parents to feel comfortable and supported with clarifying information whenever uncertainty arose.

New parents were positive about several aspects of the home environment. Many of them found it more comfortable to stay in their home environment, as long as it had a relaxed atmosphere. They felt that they had more control and could more easily follow their routines and roles in a quiet home environment compared to when they were in hospital. New parents who felt prepared for early discharge after giving birth expressed that a quiet home environment helped them to relate better to their child. Others reported that being at home enabled them to

establish a circadian rhythm around the care of their newborn baby. They were also able to learn the baby's signals at their own pace. Moreover, at home, the parental roles became more apparent, as did the feeling of being a family, with the opportunity to focus on each other's needs and support each other. Many reported an increase in self-confidence and feelings of competence. In summary, a home environment with access to digital support when needed helped parents feel that they could safely handle their new role.

### 3.4 Adequate substitute for physical meetings

As noted earlier, several new parents shared their experiences with other parents, and digital meetings were a good substitute for physical meetings [24–31]. In some studies, the new parents had limited access to digital support and most of them wanted more. Conversations via webcam helped participants to communicate more effectively because the camera enabled midwives to see what parents were talking about and thus understand the issues at hand more efficiently than when they were only described orally. After trying out various digital support systems, most new parents concluded that these were better alternatives than staying longer in hospital. As mentioned above, the majority expressed that technology was a lifeline to professional support around the clock. Some felt that even automated answers constituted a good form of support, especially those that suggested practical solutions; these helped parents avoid unnecessary hospital visits. Parents also reported having better access to healthcare professionals through the app than when they were in hospital. Many also found it easier to ask more questions in the app than in face-to-face meetings.

### 4.0 Discussion

The results from the present study highlight the new parents' experience based on sustainable and flexible digital support for parents discharged early after childbirth [24–31]. Such support is especially important for new parents, who may feel that they lack confidence in their ability to care for a newborn. Our findings are supported by Wilson et al.'s (2021) study, which highlights the healthcare sector's suitability for digital follow-up of discharged patients. The healthcare sector can improve health outcomes using digital support for patients. Digital solutions make it possible distribution of resources efficiently and fairly. Digital health interventions are context-specific and a constantly changing process. [33]. Digital follow-up makes it possible to reach a greater number of people, provide increased service coverage, and reduce healthcare costs. Indeed, Awad et al. argue that digitised support for patients in the home environment is one of the most promising developments in modern care when the technology is strategic, well thought out, and adapted to individual needs [34]. Moreover, digital devices enable rapid diagnosis, individualised treatment, and interventions such as precision surgery, symptom monitoring counselling, rehabilitation, or targeted drug delivery. An excellent example of a focus on digital support is in medical logistic are drones that can deliver necessary treatments to remote areas, collect samples, and even provide emergency assistance [11]. In maternity care, telemedicine platforms have been shown to be clinically effective and to improve patient satisfaction [35].

This review indicated that new parents were empowered when they had more knowledge regarding digital platforms than the health professionals [24–31]. This is supported by Alsem et al., who found improved parental satisfaction when parents used digital tools to prepare for consultations with healthcare professionals [36]. Alsem et al. report positive experiences of personal and interactional factors determining empowerment, creating conditions for an equal doctor–parent relationship [36]. Moreover, while they found that digital tools proved helpful for parents to explore their needs and find information, they conclude that more

research is needed to support autonomy in consultation. Salonen et al. highlight that the new generation of patients are accustomed to using the latest digital media and technological innovations [37–39]. These patients find that healthcare professionals are not always well-trained in using digital platforms [40]. This finding supports the creation of more technological solutions that meet the needs of parents in their first year postpartum [40–42].

Parents living in remote areas or lacking the support of relatives or friends are often reliant on telemedicine as a substitute for physical meetings [8]. As the COVID-19 pandemic has heightened concerns regarding infection, this has led to early hospital discharge and parental hesitancy towards in-person hospital or health centre visits and being physically seen by a healthcare professional, increasing the need for virtual consultations [15,43,44]. These have proven especially vital during lockdowns and in instances where parents have COVID-19 [45]. It was shown that the expertise obtained from existing telemedicine facilities operating in remote and rural areas has helped to shape digital follow-up after postnatal discharge [46].

Reddy et al. review has shown that most new parents have digital literacy, likely driven by the internet and the rapid increase in information and communication technology [47]. Around the world, digital health interventions are increasingly being adopted to address various public health issues. However, for users to understand, process, and act on health-related information, six literacy skills are necessary (according to the eHealth competence model): traditional, health, information, science, media, and computer literacy [48]. In the studies we reviewed, despite parents' willingness to support healthcare professionals in using digital solutions, healthcare professionals nevertheless required training in this new method of patient communication. Results from a study by Kuek and Hakkennes [49] indicate that healthcare professionals who had low digital literacy levels experienced anxiety using digital technology. This situation affected the quality of care the patients were offered. Thus, any new digital solutions should first be tested in properly designed trials to assess the level of training needed both by the healthcare professionals and the beneficiary [49].

The present review also shows that while digital solutions can empower parents newly discharged from the hospital, they can also change how care is provided. A previous study discusses how the concept of care changes when, for instance, digital support is introduced as part of the care-giving process [50,51]. Although the digital solutions explored in this review are mainly limited to apps and videoconferencing. It seems that digital solutions may represent a compromise between having a home visit from a healthcare professional and receiving no help whatsoever. However, the parents' autonomy, and whether they are empowered by digital tools, should be contextualised [52].

Our review has shown that digital solutions seems to benefit the follow-up of new parents after postnatal discharge, including by giving them 24-hour support. The digital solutions examined were mainly mobile applications, such as the mobile health app (mHealth) for postpartum care and other telemedicine solutions (e.g., videoconferencing) [53]. In addition, mHealth interventions have been shown to be effective in improving maternal and neonatal service utilisation [41,54]. These digital solutions appear to serve their purpose when care receivers wish to retrieve information by themselves or need to get in touch with healthcare professionals. In addition, the solutions seem free of potential ethical dilemmas, such as those entailed by videoconferencing solutions which can be experienced as an intrusion of privacy due to the use of cameras.

Upon discharge from hospital, new parents–even those with high digital literacy–may find it challenging or overwhelming to use digital tools whilst also caring for a newborn. Indeed, telemedicine has proven to be important for women with low socioeconomic status, particularly in relation to addressing postnatal mental health [55].

In the future, more advanced platforms or technical solutions will be available, such as social and assistive robots. Robots could be used to assist with practical tasks within the home (e.g., cleaning, and collecting and transporting items within the home) and for eHealth purposes (e.g., getting in touch with a healthcare professional or giving advice to new parents). Such robots have been tested with older adults [56–58] or other Social and Assistive Robots (SARs) [59]. However, using SARs to support new parents following discharge remains understudied and represents an area for further research. Newly discharged parents often need help with practical tasks around the home so they can focus on their newborn; some of these can potentially already be allocated to robots [50] as they do not involve direct, intimate parent–robot interaction. On the other hand, tasks that involve more interaction between the robot and the newly discharged parents, like moving and bringing items, giving advice, and helping parents get in touch with healthcare professionals, may be interesting to investigate further. However, privacy, safety, and security issues may arise depending on how advanced the robot is and what equipment is used (e.g., cameras and sensors) [60]. Rigorous pilot testing and proper clinical trials are therefore needed before these digital applications are offered to new parents discharged early from hospital.

## Conclusion

Personal counselling is an important component of care in the postpartum period, when new parents have issues specific to the child or mother (including postpartum mental health). A range of digital solutions are suitable in this situation, but a key feature of all such solutions is 24-hour access to information. In a home environment, digital communication gives parents a sense of security and emotional satisfaction; in contrast, problems can arise if parents lack access to digital resources after early discharge. The digital follow-up may be a great asset for new parents, as it can help to reliably inform them and give them the necessary support.

## Supporting information

**S1 PRISMA Checklist. PRISMA 2020 Checklist.**
(DOCX)

## Author Contributions

**Conceptualization:** Zada Pajalic.

**Formal analysis:** Zada Pajalic, Babak Asadi-Azarbaijani.

**Investigation:** Zada Pajalic.

**Methodology:** Zada Pajalic.

**Validation:** Zada Pajalic.

**Writing – original draft:** Zada Pajalic, Alona Rauckiene, Grethe Savosnick, Irena Bartels, Diana Saplacan, Sigríður Sía Jónsdóttir, Babak Asadi-Azarbaijani.

**Writing – review & editing:** Zada Pajalic, Alona Rauckiene, Grethe Savosnick, Irena Bartels, Jean Calleja-Agius, Diana Saplacan, Sigríður Sía Jónsdóttir, Babak Asadi-Azarbaijani.

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
