## [Decision Letter · Decision Letter 0]

28 Mar 2023

PDIG-D-23-00068

Digital solutions to follow up on discharged new parents

a systematic literature review

PLOS Digital Health

Dear Dr. Pajalic,

Thank you for submitting your manuscript to PLOS Digital Health. After careful consideration, we feel that it has merit but does not fully meet PLOS Digital Health's publication criteria as it currently stands. Therefore, we invite you to submit a revised version of the manuscript that addresses the points raised during the review process.

Please submit your revised manuscript within 60 days May 27 2023 11:59PM. If you will need more time than this to complete your revisions, please reply to this message or contact the journal office at digitalhealth@plos.org. Please include the following items when submitting your revised manuscript:

We look forward to receiving your revised manuscript.

Kind regards,

Haleh Ayatollahi

Section Editor

PLOS Digital Health

Journal Requirements:

Additional Editor Comments (if provided):

The manuscript was interesting. Please consider the following points in your revised manuscript.

1- Please choose appropriate keywords based on the MeSH terms.

2- Please merge the aim of the study with the rest of the introduction section. Moreover, please justify why conducting a SLR was necessary.

3- Please make sure that the structure of your manuscript (headings and subheadings) has been organized based on the PRISMA checklist. Please upload the completed check list along with your revised manuscript.

4- In the methods section, please justify why only qualitative studies were included.

5- Please explain why databases such as Scopus and Web of Knowledge were not used while these are the main databases for conducting a SLR.

6- Is there any reason for choosing the timeline between 2015 and 2022?

7- Please add a list of search strategies for all databases as an Appendix.

8- In Figure 1, excluding papers after reading the full text, the authors excluded 7 papers indicating that “7 had a low CASP assessment”. This needs to be expanded in the methods section and please support it with a reference. Moreover, in this box, reasons other than the CASP score need to be reported.

9- Table 3 is a bit difficult to follow. Could you please remove or summarize it?

10- Please re-check the captions of the Figures and Tables. In my opinion, e.g. Figure 3 is a Table not a Figure.

Reviewers' comments:

Reviewer's Responses to Questions

**Comments to the Author**

1. Does this manuscript meet PLOS Digital Health’s publication criteria? Is the manuscript technically sound, and do the data support the conclusions? The manuscript must describe methodologically and ethically rigorous research with conclusions that are appropriately drawn based on the data presented.

Reviewer #1: Yes

Reviewer #2: Partly

2. Has the statistical analysis been performed appropriately and rigorously?

Reviewer #1: Yes

Reviewer #2: No

3. Have the authors made all data underlying the findings in their manuscript fully available (please refer to the Data Availability Statement at the start of the manuscript PDF file)?

Reviewer #1: Yes

Reviewer #2: Yes

4. Is the manuscript presented in an intelligible fashion and written in standard English?

Reviewer #1: Yes

Reviewer #2: No

5. Review Comments to the Author

Reviewer #1: Summary

Most countries globally are now shifting to digital platforms in terms of health service delivery. The study focuses on digital solutions to follow up on newly-discharged parents and its effect on the parents in terms of sense of security. It was found that a range of digital solutions are suitable in this situation, but a key feature of all such solutions is 24-hour access to information. Digital communication gives parents a sense of security and emotional satisfaction; in contrast, problems can arise if parents lack access to digital resources after early discharge.

Strengths: the discussion was well-written; this study will help institutions and clinics in improving the services for parents who are newly-discharged 

Weaknesses: objective (not aligned to the results and discussion, are there other terms we can use to replace “sense of security?” What about “patient satisfaction”?), lack of supporting references/ in text citations in the results section

Evidence and examples

Major Issues:

1) Objectives: “This systematic review aims to integrate available research results that describe parents’ perceived sense of security when health and care providers used telemedicine as a platform for follow-up.” The main objective of the study is not parallel to the results and discussion. Sense of security is too vague and does not reflect the themes stated in the introduction. 

2) Methodology: How was “sense of security” assessed/measured? Was it uniform among the studies that were selected? 

3) Whole Results section - Please put specific in-text citations or supporting references for the sentences in the Results section. The references were only placed as a range “[24-31]” at the start of every section. It would be better to place the specific reference/s after every statement or so. This would help us readers check the references faster in relation to the claims in the results. 

4) Results - Flexible Digital Support: “Various digital platforms provided a combination of asynchronous and synchronous methods to facilitate communication with parents out of hospital. Asynchronous communication offered an accessible way to obtain help after early discharge from maternity units.” Maybe you can discuss and expound more on the asynchronous and synchronous methods used in the studies. What asynchronous methods were used? What synchronous methods were used? Are all of them effective and useful for follow ups?

5) Discussion: The discussion was well-written but it would be better to align it to the objective and results. Emphasize the answer to your objective. Emphasize on discussing your results.

Minor Issues:

1) Typographical error: Please add a period at the end of the sentence in the objective “This systematic review aims to integrate available research results that describe parents’ perceived sense of security when health and care providers used telemedicine as a platform for follow-up”

Reviewer #2: Abstract-

New parents or new mothers? The authors refer to these interchangeably.

Introduction-

Sentence is unclear. Suggestion:

Parity, birth-related factors, and psychosocial predictors factors indicate when it is appropriate for a new mother to be discharged and involved in the organisation of to postnatal care

Define ‘affinity’.

Only fathers? Or partners of any gender?

Repetition: There is an international trend in welfare countries with social welfare policies

Informal tone: ‘…but is about two days or fewer’ (Suggest- approximately)

‘…the provision of continuous care’ – define continuous and the context

‘perceived sense of security’ – this seems too subjective – suggest: perceptions of support

Inclusion criteria-

Refers to ALL types of telemedicine but no reference is made to the exclusion of terms including e-health, m-health or social media

Abstract/introduction refer to follow up including breastfeeding and infant feeding, and to midwives: please justify why these are then excluded from the search terms.

Please justify and discuss implications of your included studies having only four lead authors. This is a very limited insight which means claims are very bold e.g. the wider literature does not support that digital support is adequate without physical postnatal care.

Results- 

Please define and describe each theme clearly in the opening sentence for each.

Also identify which study contributed to each theme within each discussion – it is not clear where there are common findings and how the review synthesises these. The content of your Overview of themes and codes is much clearer about the differentiations than your writing. (NB This is listed as a figure when it is a table).

For example which study do you refer to here, and is it an isolated finding or a common one?: In one of the studies, the visual contact established via videoconference facilitated a warm relationship between parents and healthcare professionals.

Be specific about the platforms identified here: ‘Various digital platforms provided a combination of asynchronous and synchronous methods to facilitate communication with parents out of hospital.’ 

All themes need tighter description/names and linking to the supporting studies within the explanations for each. Content overlaps between each theme. E.g.

Flexibility and convenience of digital support is a clearer name for theme 1

Digital literacy – please remove the references to security and safety here – there is too much crossover with safety themes. Fully explain the implications of digital literacy on accessing this support if this theme has been identified.

Safe parenting – implies parenting IS safe rather than parents feeling safe. Suggest this theme is renamed to reflect this, and the inclusion of the significance of being at home as a finding. 

Adequate substitute for physical meetings – this finding is not explored adequately or linked to the studies and is not supported by the wider literature on postnatal care/ ehealth. Please justify with the evidence form the included studies.

Discussion-

Our findings are supported by Wilson et al.’s study – formalise sentence structure and place reference within the sentence not at the end throughout. Also expand – how does this evidence support your findings and why is it not included in your review?

Please clarify the relevance of this: In medical logistics, drones can deliver

necessary treatments to remote areas, collect samples, and even provide emergency assistance

Do not start a sentence with ‘And’ : And in maternity care, telemedicine platforms have been shown to be clinically effective and to improve patient satisfaction [35].

Specifically, clinically effective how?

This sentence is unclear:

The present study indicated that new parents were empowered when they had more knowledge of and facility with digital platforms than the health professionals

The present study? Do you mean this review?

What does more knowledge and facility mean? It implies parents felt empowered if they were better at using digital platforms than professionals?

Please link your claims by referencing the studies you reviewed.

This observation increases patient autonomy, which in turn affects the care

relationship [40]. – How? In what ways? Please be specific and synthesise these findings with the conclusions you draw from your review.

Parents living in remote areas or lacking the support of relatives or friends are often reliant on telemedicine as a substitute for physical meetings. -Please reference all such points and link to your findings.

The expertise obtained from existing telemedicine facilities operating in remote and rural areas has helped to shape digital follow-up after postnatal discharge [46]. – This is a paragraph which needs expanding and linking to the specific findings of your review – how does this evidence impact that you have reviewed?

Your discussion appears to focus more on other studies than those you have reviewed, and no links are made to these finidngs.

This review has shown that most new parents have digital literacy, likely driven by the internet and the rapid increase in information and communication technology – this is an overly bold statement based on very few studies with even fewer authors. Consider ‘many new parents may….’

according to the eHealth competence model) – please define and expand on this and its use in exploring your findings.

how the concept of care changes when, for instance, robots are introduced as part of

the care-giving process – define and expand – what concept of care? How does it change it? How does this impact parents/ the potential of online support?

it should be noted that the care received after discharge may be experienced differently than a home visit from a healthcare professional. – on what ways may it be experienced differently? What impact may it have (other reviews have noted the importance of concurrent support across settings)

Following a recent study describing autonomy when interacting or using digital tools as relational and situated [52], autonomy needs to be seen in context and in relation to the users’ situated abilities – define and explain these terms and the relevance of them to parents.

However, one aspect that has not been addressed is whether new parents experience the use of digital solutions differently. Differently to whom?

Upon discharge from hospital, new parents – even those with high digital literacy – may find it challenging or overwhelming to use digital tools whilst also caring for a newborn. Reference your sources.

 Indeed, telemedicine has proven to be important for women with low socioeconomic status, particularly in relation to addressing postnatal mental health [55]. How does this relate to the previous point?

In the future, more advanced platforms or technical solutions will be available, such as social

and assistive robots. Robots could be used to assist with practical tasks within the home (e.g.,cleaning, and collecting and transporting items within the home) and for eHealth purposes (e.g., getting in touch with a healthcare professional or giving advice to new parents). Such robots have been tested with older adults – see for instance, the use of the Giraff robot [56-58] or other social and assistive robots (SARs) [59]. However, using SARs to support new parents following discharge remains understudied and represents an area for further research. Newly discharged parents often need help with practical tasks around the home so they can focus on their newborn; some of these can potentially already be allocated to robots [50] as they do not involve direct, intimate parent–robot interaction. On the other hand, tasks that involve more 16 interaction between the robot and the newly discharged parents, like moving and bringing items, giving advice, and helping parents get in touch with healthcare professionals, may be interesting to investigate further. However, privacy, safety, and security issues may arise depending on how advanced the robot is and what equipment is used (e.g., cameras and sensors) [60]. Rigorous pilot testing and proper clinical trials are therefore needed before these digital applications are offered to new parents discharged early from hospital.

The above seems irrelevant to your findings and the scope of your review. You need to properly define the parameters of ‘postnatal care/support’. You have also not explained the terms discussed e.g Giraff robot and what they do. Please remove or clearly explain the links to your findings.

Implications for practice

After early home discharge, new parents should have reliable, 24-hour access to information

related to newborn care, breastfeeding, and parenthood. - You have not clearly identified how and which of your themes have demonstrated this.

This is especially important when supporting breastfeeding while also ensuring security and cultural appropriateness. - Other than in your abstract you have not referred anywhere to breastfeeding support, nor included this in your search terms – this is therefore not a claim supported b your review.

Today’s new parents tend to be digitally literate and are thus very likely to make use of telemedicine. – again there are no figures or references to the literature to justify this claim. I suggest Statista.

Conclusion

in contrast, problems can arise if parents lack access to digital resources after early discharge. – You have not demonstrated this to be the case with your findings or links to wider evidence.

Finally, digital literacy is important for both healthcare professionals and patients receiving care

through telemedicine. – You have not provided sources or reference to digital literacy amongst professionals elsewhere

This area needs to be developed in the healthcare system, especially among midwives, who mainly provide this support in the context of postnatal follow-up care. Again you need to explore the literature on midwives and digital support to claim this. See Morse & Brown, 2021 and 2022.

Additionally: you have provided no limitations to your review of which there are several, including the limited number of included papers, the search terms which do not include breastfeeding or social media or midwives. Please add a limitations section.

6. PLOS authors have the option to publish the peer review history of their article (what does this mean?). If published, this will include your full peer review and any attached files.

**Do you want your identity to be public for this peer review?** For information about this choice, including consent withdrawal, please see our Privacy Policy.

Reviewer #1: Yes: Arianne Justine Obeles

Reviewer #2: Yes: Dr Holly Morse

---

## [Decision Letter · Decision Letter 1]

30 May 2023

PDIG-D-23-00068R1

Digital solutions to follow up on discharged new parentsa systematic literature review

PLOS Digital Health

Dear Dr. Pajalic,

Thank you for submitting your manuscript to PLOS Digital Health. After careful consideration, we feel that it has merit but does not fully meet PLOS Digital Health's publication criteria as it currently stands. Therefore, we invite you to submit a revised version of the manuscript that addresses the points raised during the review process.

Please submit your revised manuscript within 30 days Jun 29 2023 11:59PM. If you will need more time than this to complete your revisions, please reply to this message or contact the journal office at digitalhealth@plos.org. Please include the following items when submitting your revised manuscript:

We look forward to receiving your revised manuscript.

Kind regards,

Haleh Ayatollahi

Section Editor

PLOS Digital Health

Journal Requirements:

Additional Editor Comments (if provided):

I appreciate the authors for their time and efforts to revise the manuscript. Please consider the following minor revision in your manuscript, too.

1- In all tables, please report the studies chronologically.

2- Please make sure that all of the submitted materials are in English.

3- Regarding the search strategies, I think it is enough if you please provide a table and put the name of each database along with the final search strategy for that database in the table.

Reviewers' comments:

Reviewer's Responses to Questions

**Comments to the Author**

1. If the authors have adequately addressed your comments raised in a previous round of review and you feel that this manuscript is now acceptable for publication, you may indicate that here to bypass the “Comments to the Author” section, enter your conflict of interest statement in the “Confidential to Editor” section, and submit your "Accept" recommendation.

Reviewer #1: All comments have been addressed

Reviewer #2: (No Response)

2. Does this manuscript meet PLOS Digital Health’s publication criteria? Is the manuscript technically sound, and do the data support the conclusions? The manuscript must describe methodologically and ethically rigorous research with conclusions that are appropriately drawn based on the data presented.

Reviewer #1: Yes

Reviewer #2: Partly

3. Has the statistical analysis been performed appropriately and rigorously?

Reviewer #1: Yes

Reviewer #2: Yes

4. Have the authors made all data underlying the findings in their manuscript fully available (please refer to the Data Availability Statement at the start of the manuscript PDF file)?

Reviewer #1: (No Response)

Reviewer #2: Yes

5. Is the manuscript presented in an intelligible fashion and written in standard English?

Reviewer #1: Yes

Reviewer #2: Yes

6. Review Comments to the Author

Reviewer #1: All comments have been addressed. Thank you for responding to the comments.

Reviewer #2: There are some comments which have not been addressed - please see attached document. The main revision required as also stated by the other reviewer is to cite the specific study each claim relates to under each theme, so the reader can identify how the reviewed work relates to the findings. I suggest looking at similar reviews using TA if how this can be done is not clear for example https://onlinelibrary.wiley.com/doi/full/10.1111/mcn.13399

7. PLOS authors have the option to publish the peer review history of their article (what does this mean?). If published, this will include your full peer review and any attached files.

**Do you want your identity to be public for this peer review?** For information about this choice, including consent withdrawal, please see our Privacy Policy. 

Reviewer #1: Yes: Arianne Justine T. Obeles

Reviewer #2: Yes: Dr Holly Morse

---

## [Editor Report · Decision Letter 2]

29 Jun 2023

PDIG-D-23-00068R2

Digital solutions to follow up on discharged new parentsa systematic literature review

PLOS Digital Health

Dear Dr. Pajalic,

Thank you for submitting your manuscript to PLOS Digital Health. After careful consideration, we feel that it has merit but does not fully meet PLOS Digital Health's publication criteria as it currently stands. Therefore, we invite you to submit a revised version of the manuscript that addresses the points raised during the review process.

Please submit your revised manuscript within 30 days Jul 29 2023 11:59PM. If you will need more time than this to complete your revisions, please reply to this message or contact the journal office at digitalhealth@plos.org. Please include the following items when submitting your revised manuscript:

We look forward to receiving your revised manuscript.

Kind regards,

Haleh Ayatollahi

Section Editor

PLOS Digital Health

Journal Requirements:

Additional Editor Comments (if provided):

The following issues still need to be addressed in your revision.

1- The title has not been edited in the submission system. Please edit it.

2- After Table 1, there is still another table with empty cells. Please remove it.

3- Tables 3 shows “Overview of themes and codes”. In the systematic reviews, we usually do not present codes.

4- Figure 3, actually it is not a Figure. It is a table which shows themes and subthemes. Again, in the systematic reviews, we usually do not present such a table, but the results section can be divided based on the themes and subthemes.

5- In terms of the search strategies, just one final search strategy for each database is enough. Search strategies for different databases can be presented in one table to be submitted as a supplementary file.
---

## [Editor Report · Decision Letter 3]

5 Jul 2023

Digital solutions to follow up on discharged new parents - a systematic literature review

PDIG-D-23-00068R3

Dear Professor Pajalic,

We are pleased to inform you that your manuscript 'Digital solutions to follow up on discharged new parents - a systematic literature review' has been provisionally accepted for publication in PLOS Digital Health.

Best regards,

Haleh Ayatollahi

Section Editor

PLOS Digital Health